# Generation of Iron-Independent Siderophore-Producing *Agaricus bisporus* through the Constitutive Expression of *hapX*

**DOI:** 10.3390/genes12050724

**Published:** 2021-05-13

**Authors:** Min-Seek Kim, Hyeon-Su Ro

**Affiliations:** Department of Bio & Medical Big Data and Research Institute of Life Sciences, Gyeongsang National University, Jinju 52828, Korea; kmstaur@gmail.com

**Keywords:** *Agaricus bisporus*, transformation, siderophore, expression

## Abstract

*Agaricus bisporus* secretes siderophore to uptake environmental iron. Siderophore secretion in *A. bisporus* was enabled only in the iron-free minimal medium due to iron repression of *hapX*, a transcriptional activator of siderophore biosynthetic genes. Aiming to produce siderophore using conventional iron-containing complex media, we constructed a recombinant strain of *A. bisporus* that escapes *hapX* gene repression. For this, the *A. bisporus*
*hapX* gene was inserted next to the glyceraldehyde 3-phosphate dehydrogenase promoter (pGPD) in a binary vector, pBGgHg, for the constitutive expression of *hapX*. Transformants of *A. bisporus* were generated using the binary vector through *Agrobacterium tumefaciens*-mediated transformation. PCR and Northern blot analyses of the chromosomal DNA of the transformants confirmed the successful integration of pGPD-*hapX* at different locations with different copy numbers. The stable integration of pGPD-*hapX* was supported by PCR analysis of chromosomal DNA obtained from the 20 passages of the transformant. The transformants constitutively over-expressed *hapX* by 3- to 5-fold and *sidD*, a key gene in the siderophore biosynthetic pathway, by 1.5- to 4-fold in mRNA levels compared to the wild-type strain (without Fe^3+^), regardless of the presence of iron. Lastly, HPLC analysis of the culture supernatants grown in minimal medium with or without Fe^3+^ ions presented a peak corresponding to iron-chelating siderophore at a retention time of 5.12 min. The siderophore concentrations of the transformant T2 in the culture supernatant were 9.3-fold (−Fe^3+^) and 8-fold (+Fe^3+^) higher than that of the wild-type *A. bisporus* grown without Fe^3+^ ions, while no siderophore was detected in the wild-type supernatant grown with Fe^3+^. The results described here demonstrate the iron-independent production of siderophore by a recombinant strain of *A. bisporus*, suggesting a new application for mushrooms through molecular biological manipulation.

## 1. Introduction

Iron is an essential trace element that combines with proteins or small organic molecules involved in various oxidation–reduction reactions in living organisms. Due to the scarcity of iron in nature, microorganisms secure iron by secreting siderophores and absorbing them back into the cells following the adsorption of environmental iron [1]. Siderophores are small organic iron chelators that preferentially bind to Fe^3+^ ions through the formation of a coordination complex. They are classified by the ligand structure as carboxylates, hydroxamates, catecholates, and mixed types [2]. Siderophores, including enterobactin (catecholate), desferrioxamine (hydroxamate), and pyoverdine (mixed type), are known to be produced by bacteria [3], whereas hydroxamate siderophores, such as ferrichromes, fusarinine C, and triacetylfusarinine C (TAFC), are the major siderophores in fungi [1,4].

The fungal hydroxamate siderophores are synthesized by the activity of the nonribosomal peptide synthetase complex (NRPS). In *Aspergillus fumigatus*, SidC is involved in the biosynthesis of ferrichromes using N^5^-acetyl-N^5^-hydroxy-l-ornithine, glycine, and serine as precursors, where SidD catalyzes the condensation of three molecules of N^5^-anhydromevalonyl-N^5^-hydroxy-l-ornithine to produce fusarinine C and TAFC [1]. Basidiomycetes also have shown to contain similar NRPSs. Sid2 and Fso1 in *Ustilago maydis* [5] and *Omphalotus olearius* [6], respectively, are NRPSs homologous to SidC in *A. fumigatus,* whereas CsNPS2 in *Ceriporiopsis subvermispora*, NPS1 in *Trametes versicolor,* and the gene product of EAU88504 in *Coprinopsis cinerea* are thought to be homologous to SidD as they show similar domain arrangement [7]. *Agaricus bisporus* is reported to produce hydroxamate siderophores, such as ferrichrome, defferri-des(diserylglycyl) ferrirhodin, and fusarinine C [8], accordingly, and putative SidC and SidD homologues are found from the genome information of *A. bisporus* as described in the present study.

The expression of genes involved in the siderophore biosynthetic pathway is tightly regulated by the transcription factor *hapX*, which positively regulates the expression of *sid* genes and represses genes involved in iron-consuming pathways under iron-starvation conditions in *A. fumigatus* [9]. The expression of *hapX* is down-regulated by another transcription factor, *sreA*, under sufficient iron conditions [10,11]. Study on iron metabolism in basidiomycetes is rare. However, *hapX* and *cir1*, a homologue of *sreA*, in *Cryptococcus neoformans* have been extensively studied in relation to virulence in humans [12,13]. Similar to the ascomycetes, *hapX* in *C. neoformans* represses iron utilization and promotes iron uptake and *cir1* expression under iron starvation conditions [12,13]. Therefore, the deprivation of iron is a prerequisite for the expression of *hapX* and thus for the production of siderophore in fungi.

Siderophores have multiple applications in agriculture as plant growth promotors and pathogen control agents, as well as in the removal of contaminated heavy metal ions [14]. In medicinal applications, siderophores are employed to treat diseases associated with iron overload [15]. Certain siderophores show antimicrobial activities against pathogenic bacteria, though siderophores mostly act as virulence factors for many pathogens [16,17,18]. Drug delivery through siderophore receptors after the formation of siderophore–drug complex may be a new strategy to deliver drugs that have permeability-mediated drug resistance [19]. Additionally, ferrichrome produced by *Lactobacillus casei* has been reported to show anticancer activity against colon cancer in a mouse model [20].

In the present study, we assess the potential of *A. bisporus*, a representative edible mushroom, as a host system for the production of biological molecules such as siderophores. However, *A. bisporus*, like other fungi, secreted siderophores only in the absence of iron, meaning that the production medium should be an iron-free minimal medium. To overcome this problem, we generated transformant strains of *A. bisporus* that express *hapX*, a master transcription factor for *sid* gene expression, using the constitutive glyceraldehyde 3–phosphate dehydrogenase promoter (pGPD).

## 2. Materials and Methods

### 2.1. Strains and Culture Conditions

*A. bisporus* NH1 was obtained from the National Institute of Horticultural and Herbal Science, Rural Development Administration (RDA; Eomsung, Korea). The mushroom strain was grown at 25 °C on compost–potato dextrose agar (C–PDA), in which PDA (Oxoid, Basingstoke, UK) was supplemented with 200 mL of compost extract per litter. For the liquid culture, compost–potato dextrose broth (C–PDB) was prepared by supplementing the compost extract (200 mL/L) with potato dextrose broth (Oxoid, Basingstoke, UK). The compost extract for the preparation of both media was prepared as follows: rice straw compost was suspended in water (100 g/L), left for a day at room temperature, autoclaved for 40 min at 121 °C, then filtered using a Miracloth (Sigma-Aldrich, St. Louis, MO, USA).

### 2.2. Construction of pBGgHg-hapX Vector

The *hapX* gene sequence in *A. bisporus* was retrieved from the MycoCosm genome database (https://mycocosm.jgi.doe.gov/Agabi_varbisH97_2/Agabi_varbisH97_2.home.html, accessed on 16 August 2019) by BLASTP analysis using *A. fumigatus* HapX protein (XP_747952.1) as a query sequence. The retrieved HapX protein contained a conserved Hap2/3/5 binding domain in front of the bZIP domain similar to *A. fumigatus* HapX protein (Appendix A). Homologous gene sequences to *A. fumigatus sidC* and *sidD* were also retrieved under the gene accession numbers NW_006267370.1 and NW_006267372.1, respectively, using a similar approach.

For the isolation of *hapX* gene, *A. bisporus* NH1 was grown in 100 mL of C–PDB for 10 days at 25 °C. The genomic DNA was isolated from the harvested mycelia using a genomic DNA extraction kit (HiGene Genomic DNA Prep kit; BIOFACT, Daejeon, Korea). The *A. bisporus hapX* gene (GenBank Accession No. XM_006454360) was amplified from the purified genomic DNA using a specific primer set (Appendix A, hapX-F and hapX-R) and a PCR premix (nPfu-Forte; Enzynomics, Daejeon, Korea). The PCR reaction consisted of the initial denaturation at 95 °C for 5 min, followed by 25 cycles of denaturation at 95 °C for 30 s, annealing at 60 °C for 30 s, and polymerization at 72 °C for 4 min, and a final extension at 72 °C for 10 min. The resulting amplicon contained *Swa*I and *Spe*I restriction sites at the 5′- and 3′-ends of the *hapX* gene provided by the primers, respectively. The *hapX* amplicon was digested by *Swa*I and *Spe*I and inserted into the corresponding restriction sites in the pBGgHg vector [21] by replacing the *eGFP* gene, generating pBGgHg-hapX (Appendix A).

### 2.3. Transformation of Agrobacterium tumefaciens AGL-1 with pBGgHg-hapX

*Agrobacterium tumefaciens* AGL-1 was grown in 10 mL of LB (peptone 10 g/L, yeast extract 5 g/L, NaCl 5 g/L) at 28 °C until the optical density at 600 nm reached 0.6. The bacterial cells were collected by centrifugation (10,000× *g*, 10 min). The cell pellet was washed with 1X TE buffer and resuspended in 2 mL of 10% LB broth. Aliquots of the competent cells were stored at −70 °C until use. For the transformation, the competent cells (250 μL) were mixed with 1 μg of pBGgHg-hapX plasmid DNA by gentle pipetting and were kept on ice for 5 min. After the incubation, the tube was placed in liquid nitrogen for 5 min and then transferred to a water bath (37 °C). After 5 min of incubation, 1 mL of LB broth was added and incubated for an additional 2 h at 28 °C with vigorous agitation. The incubated cells were collected by centrifugation and were suspended in 0.2 mL LB broth. The suspension was spread on LB agar containing 50 mg/L of kanamycin. The plate was incubated for 2 days at 28 °C to obtain *A. tumefaciens* AGL-1 transformed with pBGgHg-hapX.

### 2.4. Agrobacterium tumefaciens-Mediated Transformation of Agaricus bisporus

*A. bisporus* was transformed by the *A. tumefaciens*-mediated transformation (ATMT) method [21] with slight modifications. For the transformation, 1 mL of freshly grown *A. tumefaciens* AGL-1 harboring pBGgHg-hapX was inoculated to 100 mL of LB broth and incubated at 28 °C with agitation (170 rpm) until the optical density at 600 nm reached 0.8. The cells were collected by centrifugation (10,000× *g*, 10 min) and were resuspended in 100 mL of induction medium (IM), containing 2 g/L glucose, 5 mL/L glycerol, 1 g/L NH_4_Cl, 0.3 g/L MgSO_4_, 0.15 g/L KCl, 0.01 g/L CaCl_2_, 2.5 mg/L FeSO_4_, 0.2 mM acetosyringone, 40 mM MES buffer (pH 5.3), and 50 mg/L kanamycin. The cell suspension was incubated for 6 h at 25 °C. For the infection of *A. tumefaciens* to *A. bisporus*, pieces of gill tissue (1 mm) were cut from the fruiting body of *A. bisporus* NH1 and were co-incubated in the bacterial culture for 15 min. The infected gill tissues were transferred onto IM agar (IM with 15 g/L agar). After 3 days of incubation at 25 °C, the gill tissues were transferred to the first selection medium (C-PDA supplemented with 150 mg/L cefotaxime, 100 mg/L kanamycin, 25 mg/L chloramphenicol, gentamycin 100 mg/L, and 30 mg/L hygromycin B). After 10 days of incubation at 25 °C, the tissues showing filamentous hyphal growth were transferred to the second selection medium (C-PDA with 50 mg/L hygromycin B). The mycelia outgrowing on the second selection medium upon prolonged incubation at 25 °C were cut out from the agar medium and were subjected to further analysis.

### 2.5. PCR and Real-Time PCR Analyses of the A. bisporus Transformants

The transformants of *A. bisporus* obtained from ATMT were analyzed by PCR using primer sets targeting an internal sequence region in the *hph* gene (731 bp; hph-F and hph-R) and a sequence region ranging from pGPD to the internal *hapX* sequence (725 bp; Pgpd-F and hapXin-R) (Figure 1a and Appendix A). The PCR conditions were the same as those described above, except for the polymerization time (30 s in this case). The expression of the *hapX* gene in the transformants was investigated by quantitative real-time PCR (qPCR) analysis. For the qPCR, 1 mL of actively grown mycelia in C-PDB were inoculated in 50 mL of minimal medium (KCl 0.2 g/L, KH_2_PO_4_ 0.14 g/L, Na_2_HPO_4_·12H_2_O 1.9 g/L, CaCl_2_·2H_2_O 0.27 g/L, MgSO_4_·7H_2_O 0.2 g/L, ZnSO_4_·7H_2_O 2 mg/L, CuSO_4_·6H_2_O 0.1 mg/L, MnSO_4_·H_2_O 0.02 mg/L, (NH_4_)_6_Mo_7_O_24_·4H_2_O 0.02 mg/L, H_3_BO_3_ 0.01 mg/L, glucose 30 g/L, yeast extract 5 g/L, pH 6.8) with or without 10 μM FeCl_3_. The culture was incubated for a week at 25 °C. The harvested mycelia were subjected to total RNA extraction. Total RNA was extracted using an RNeasy Plant Mini Kit (Qiagen, Hilden, Germany) from the mycelial powder (0.1 g), which was prepared by grinding with a pestle and mortar after freezing in liquid nitrogen. The extracted RNA was subjected to cDNA synthesis using a TOPscript cDNA Synthesis Kit (Enzynomics, Daejeon, Korea), followed by qPCR using FastStart Universal SYBR Green Master (Sigma-Aldrich) and Lightcycler Nano (Roche, Germany). The primers used for qPCR are shown in Appendix A. The qPCR results were analyzed using the expression of the β-tubulin gene as a reference. Relative gene repression was calculated using the 2^−∆∆Cq^ value. All data were obtained in triplicate from three independent experiments. The statistical significance between data sets was analyzed using a one-way ANOVA test.

### 2.6. Southern Blot Analysis with DIG-Labeling 

Genomic DNA (20 μg) was digested with *Spe*I and *Age*I for 24 h at 37 °C. The digested samples were subjected to agarose gel electrophoresis in 0.8% agarose gel (15 cm × 15 cm) at 30 V for 18 h, followed by 50 V for 3 h. A digoxigenin (DIG)-labeled DNA marker (DNA molecular weight marker III, Roche, Germany) was run together for the size analysis. The DNA fragments in the agarose gel were transferred onto a nylon membrane (Amersham hybond^TM^-N^+^ nylon membrane, GE Healthcare, Chicago, IL, USA) using standard blotting methods. After the transfer, the DNA fragments on the membrane were cross-linked using CL-1000 Ultraviolet Crosslinker (Spectrum chemical, New Brunswick, NJ, USA). The nylon membrane was subjected to hybridization with a 10 ng/mL DIG-labeled probe specific to the *hapX* gene. The DIG-labeled *hapX* probe was generated by PCR using hapX-probe-fwd and hapX-probe-rev primers (Appendix A) followed by DIG labeling using a labeling kit (Random Primed DNA Labeling Kit, Roche, Rotkreuz, Switzerland). The DIG-labeled DNA was visualized by the NBT/BCIP reaction after binding with 150 mU/mL of Anti-DIG-AP conjugate (Roche, Rotkreuz, Switzerland).

### 2.7. HPLC Analysis

The mycelia of *A. bisporus* were grown in C–PDB for 10 days and were harvested by centrifugation (1000× *g*, 10 min). The collected mycelia were resuspended in minimal medium and further incubated for 3 days. The culture broth was obtained through filtration with a Miracloth (Sigma-Aldrich, St. Louis, MO, USA) followed by 3 M filter paper. Final 1 mM of FeCl_3_ was added to the filtrate to convert siderophores to iron-bound form. The treated filtrate (2 mL) was subjected to a Sep-Pak C18 cartridge (Waters, Milford, MA, USA). The cartridge was washed with 10 mL of deionized water, and then the bound siderophore was eluted by 5 mL of methanol. The eluate was dried by a vacuum evaporator. The dried sample was dissolved in 0.5 mL of deionized water. The obtained sample (20 μL) was subjected to HPLC analysis using an HPLC system (HP1050; Hewlett-Packard, Palo Alto, CA, USA) equipped with an HC-C18(2) column (150 × 4.6 mm; Agilent, Santa Clara, CA, USA). Isocratic elution was performed with 0.1% triflouroacetic acid as a mobile phase at a flow rate of 0.6 mL/min. The chromatogram was monitored at 214 nm.

### 2.8. Determination of Siderophore Activity by Chrome Azurol-S Assay

Siderophore in the sample solution was determined using a modified chrome azurol-S (CAS) assay [22]. For the formulation of the CAS reagent, 0.75 mL of 2 mM CAS was mixed with 1 mL of 10 mM FeCl_3_ (in 10 mM HCl) and 0.6 mL of 10 mM hexadecyltrimethylammonium (HDTMA). The final volume of the mixture was brought up to 10 mL with deionized water after the addition of 5 mL of 1 M MES (pH5.6) to make the CAS reagent. For the determination of siderophore, the peak fraction from the HPLC analysis (0.6 mL) was treated with 60 μL of 3% (*w*/*v*) 8-hydroxyquinoline (Sigma-Aldrich, St. Louis, MO, USA) in chloroform. The chloroform layer was removed after centrifugation (10,000× *g*, 5 min). The aqueous layer (0.5 mL) was mixed with the CAS reagent (0.5 mL) and 10 μL of 0.2 M sulfosalicylic acid. The reaction mixture was incubated for 1 h at 25 °C. The decrease in the absorbance at 650 nm (A_650_) was monitored using a UV-vis spectrophotometer. The relative production of siderophore was calculated using the following equation.
(1)$$\mathrm{Relative}\;\mathrm{production}=\left(\frac{A^{\max}-A^{\mathrm{sample}}}{A^{\max}-A^{\min}}\right)\times 100$$
where A^max^ stands for the A_650_ value without siderophore and A^min^ for the A_650_ value at the saturated siderophore concentration. A^sample^ represents the A_650_ value in the presence of the siderophore sample.

## 3. Results

### 3.1. Generation of Transformants for the Constitutive Expression of hapX

After ATMT with pBGgHg-hapX, a total of 172 mycelial isolates grown out of the gill tissues on the selection medium were examined by PCR with primer sets targeting pGPD-*hapX* and *hph* (Figure 1a). As a result, 42 isolates (24.4%) were found to contain pGPD-*hapX* and *hph* as an indication of successful integration (Figure 1b). Next, the genomic DNA of 10 randomly selected transformants was subjected to Southern blot analysis to further confirm the genomic integration of pGPD-*hapX*. The DIG-labeled probe targeting the *hapX* gene detected a DNA band containing an endogenous *hapX* gene with a size of 15.5 kb (Figure 1c, arrow), as well as an integrated *hapX* gene of variable size in the transformants (Figure 1c). The transformants carried at least one additional copy of the *hapX* gene. The transformants, including T4, T7, T13, T18, T22, T23, and T24, were found to contain a single additional *hapX* gene of variable size in the genomic DNA fragment, whereas T1 and T15 had two additional copies of the *hapX* gene. The transformant T2 had four copies of the *hapX* gene, of which three were from the genomic integration of pGPD-*hapX*.

**Figure 1 genes-12-00724-f001:**
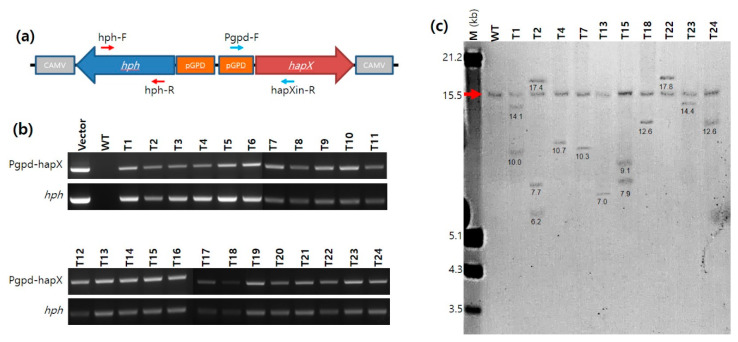
The transformation of *Agaricus bisporus*. (**a**) Gene arrangement of the integrating unit in pBGgHg-hapX. Red arrows and blue arrows indicate the positions of primers for the detection of *hph* and pGPD-*hapX*, respectively. CAMVs are CAMV poly(A) signals residing in front LB and RB of pBGgHg-hapX. (**b**) PCR analysis of the transformants. The amplicons were approximately 500 bp for both targets. (**c**) Southern blot analysis of the transformants. Genomic DNA (20 μg) was isolated from liquid cultures, digested with *Age*I and *Sac*I, and probed with an ~675 bp DIG-labeled *hapX* gene sequence. Lanes; M, DNA molecular size marker (kb); WT, untransformed *A. bisporus*; T1 to T24, transformants. The red arrow indicates the original *hapX* gene fragment. Numbers under the DNA bands indicate the sizes of genomic DNA fragments.

Successive transfer of the transformants on C-PDA (hygromycin B-free) medium was performed to investigate the genetic stability of the integrated pGPD-*hapX* and *hph* genes in the host genome. The growing edge of mycelia was successively transferred to fresh C-PDA every 3 weeks. After 20 successive transfers, the presence of pGPD-*hapX* and *hph* genes in the genomic DNA of the transformants was examined by PCR with the primer sets described above. The PCR analysis revealed that the integrated DNA was stably maintained (Appendix A). There was no difference in the growth characteristics between the wild-type (WT) and the transformants even after the twenty successive transfers (Appendix A).

### 3.2. Effect of Iron on the mRNA Expression of hapX

The expression of *hapX* in the transformants was investigated by qPCR using total RNAs extracted from mycelial cells grown in minimal medium with or without 10 μM Fe^3+^. The *hapX* gene expression in the WT was repressed in the presence of Fe^3+^ (Figure 2a), together with the repression of *sidD*, a downstream component of the *hapX* gene (Figure 2b). Removal of Fe^3+^ resulted in 6.1- and 7.9-fold increases in the *hapX* and *sidD* expression, respectively. In contrast, the presence of Fe^3+^ no longer repressed the expression of both genes in the four selected pGPD-*hapX* transformants because of the constitutive expression by the pGPD promoter. The transformants showed approximately 3- to 5-fold overexpression of *hapX* and 1.5- to 4-fold overexpression of *sidD*, independently of Fe^3+^, when compared with the corresponding gene expression in the WT grown in iron-free medium (Figure 2). Removal of Fe^3+^ resulted in slight but statistically insignificant changes in the gene expression levels among the transformants, except for T2, which showed 1.2- and 1.4-fold greater expression in *hapX* and *sidD*, respectively, in the absence of Fe^3+^. The multiple integrations of pGPD-*hapX* into the genomic DNA appeared to influence the *hapX* and *sidD* expression; however, the copy number did not linearly correlate with the gene expression level. The transformant T2, which contained three additional transgenic copies of the *hapX* gene, showed only 1.3-fold more *hapX* gene expression than T15, which had two additional copies of the *hapX* gene. Moreover, the expression level of *sidD* in T2 was almost the same as that in T15. Notably, the *sidD* expression in T1, which has two additional *hapX* copies, was lower than that in T15.

### 3.3. Production of Siderophore by the pGPD-hapX Transformants

The culture supernatants of WT and T2 strains that were grown in minimal medium with or without Fe^3+^ were subjected to HPLC analysis to compare siderophore production. The wild-type strain showed only a small peak corresponding to siderophore at a retention time of 5.12 min from the culture broth without Fe^3+^ (Figure 3). HPLC analysis of the T2 strain revealed the presence of the same siderophore peak from both iron-free [T2(−Fe)] and iron-included [T2(+Fe)] culture supernatants of T2 (Figure 3). However, the concentration of siderophore produced from T2 was 9.3-fold and 8.0-fold higher for iron-free and iron-included supernatants, respectively, than that from the wild-type, as compared by the peak area. The siderophore production was comparable to the expressed mRNA level of the *hapX* gene.

The iron-chelating activity of the siderophore in the peak fraction of HPLC was investigated by CAS assay. Deprivation of Fe^3+^ in the CAS–Fe^3+^ complex could be monitored by the decrease in the absorbance at 650 nm, as observed in the EDTA treatment (Appendix A). Treatment of the peak fraction from WT(−Fe) resulted in very little change in the CAS–Fe^3+^ spectrum because of low siderophore concentration (Figure 3c). However, the peak fraction from either T2(−Fe) or T2(+Fe) showed high Fe^3+^-chelating activity by removing the absorption peak at 650 nm, as an indication of high siderophore production.

Next, we investigated the effects of pH and temperature on the mycelial growth and siderophore production of the WT and T2 in liquid culture. Both the WT and T2 showed better growth at pH 6–7, although the mycelial growth was not affected greatly in the range of pH 5–7.5 (Figure 4). However, the siderophore production was highly dependent on pH, having the highest production level near a pH of 7.0 for both WT and T2. The siderophore production of T2 was approximately 10-fold higher than that of the WT in the investigated pH range. Both the WT and T2 showed optimal growth at 25~30 °C with maximal siderophore production within the temperature range. Mycelial growth below or above this temperature range was decreased in both strains. It is notable that the iron-chelating activity observed in WT may come from iron-chelating metabolites, such as oxalate and citrate.

## 4. Discussion

Mushrooms mostly belonging to the phylum Basidiomycota are an important group of fungi due to their roles as decomposers in the ecosystem. Some of them, including *A. bisporus*, *Lentinula edodes*, *Pleurotus ostreatus*, and *Flammulina velutipes*, are mainly consumed as nutritious foods, while others, such as *Ganoderma lucidum*, *Phellinus linteus*, *Inonotus obliquus*, and *Trametes versicolor*, are used as sources of biologically active compounds [23]. Although they have a long history of human consumption, there have been few studies on the genetic (metabolic) engineering of basidiomycetes in order to use them as biological hosts to produce small organic compounds, despite the fact that they are equipped well with various metabolic pathways and diversified by their gene duplication and horizontal gene transfer [24].

*A. bisporus* is one of the best-studied commercial mushrooms in genetics and molecular biology [21,25,26,27], making it a good model system for the genetic engineering of mushrooms. In the present study, we demonstrated the successful transformation of *A. bisporus* to produce siderophore independently of iron, as normally, *A. bisporus* only produces siderophore in the absence of iron to acquire environmental iron. In filamentous fungi, *sid* genes involved in the siderophore biosynthesis are under the control of the main transcriptional activator, *hapX*, the expression of which is negatively controlled by ferrous ion [9]. For the constitutive expression of *hapX*, we isolated a *hapX* homolog and GPD promoter (pGPD) from the *A. bisporus* chromosomal DNA and performed genomic integration of pGPD-*hapX* through *A. tumefaciens*-mediated transformation. The transformation efficiency (24%) was comparable to that which was previously reported (30–40%) [21]. Southern blot analysis revealed that single-copy integration of pGPD-*hapX* was prevalent; however, the transformants with 2–3 copies of pGPD-*hapX* integration were also observed, similar to a previous report on pBGgHg [21]. The multiple copies observed in certain transformants in this study do not imply multiple integrations to the chromosomes of a certain nucleus because *A. bisporus* can have multiple nuclei in the cytoplasm. Nonetheless, in contrast to this previous report, which failed to detect the expression of the *eGFP* gene [21], we observed the functional expression of *hapX* at the mRNA level (Figure 2) and also at the protein level, as deduced from the constitutive production of siderophore (Figure 3). This is conceivable because *hapX* originates from *A. bisporus* itself, thereby enabling normal mRNA processing and the use of host codons, in contrast to heterologous *eGFP* [21]. Additionally, we showed that integrated pGPD-*hapX* is maintained stably through successive transfers in the absence of selective pressure.

*A. bisporus* has been known to produce three hydroxamate siderophores, including ferrichrome (FC), defferri-des(diserylglycyl) ferrirhodin (DDF), and fusarinine C (FsC), at the ratios of 10%, 30%, and 60%, respectively [8]. The first two siderophores are known to be synthesized by the NRPS activity of the *sidC* gene product using N^5^-acetyl-N^5^-OH-L-Orn, glycine, and serine as precursors, whereas FsC is a condensation product comprised of three molecules of N^5^-anhydromevalonyl-N^5^-OH-L-Orn catalyzed by the NRPS encoded by *sidD* [1]. The true nature of the siderophore produced from the pGPD-*hapX* transformants was not identified; however, our RT-PCR analysis showed that only the *sidD* gene was transcribed in a *hapX*-dependent manner, while the expression of *sidC* was not observed (data not shown), suggesting that the siderophore produced in the present study is FsC.

Since mushrooms have mostly been consumed as fresh foods or medicines, research on their molecular genetic modification is very rare. However, considering the fact that mushrooms produce various organic molecules in their different developmental stages, they have great potential as host systems for the production of useful organic compounds. In this regard, our research provides a good example of how a common mushroom can be transformed into a producer of valuable organic compounds using genetic manipulation.

## 5. Conclusions

Iron inhibits siderophore biosynthesis in *A. bisporus* by repressing the *hapX* transcription factor. Using *A. bisporus* transformants that constitutively express *hapX* through the GPD promoter, we demonstrate the iron-independent production of siderophore. The constitutive expression of *hapX* leads to the expression of *sidD*, an NRPS gene, which results in the production of siderophore in the iron-containing complex medium.

## Figures and Tables

**Figure 2 genes-12-00724-f002:**
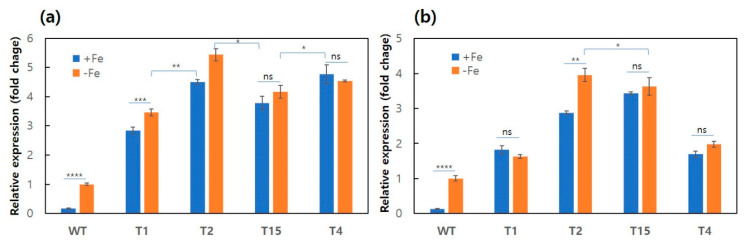
mRNA expression level comparison of transformants. Relative RNA expression of *hapX* (**a**) and *sidD* (**b**) in the wild-type (WT) and the transformants T1, T2, T3, and T4. Each strain was treated with 10 μM FeCl_3_ (+Fe) or was held in iron-free (−Fe) conditions for a 24 h incubation period. Error bars indicate the standard deviations of the means. The statistical significance of the mean difference between samples is indicated on the bar using asterisks (* for *p* ≤ 0.05, ** for *p* ≤ 0.01, *** for *p* ≤ 0.001, **** for *p* ≤ 0.0001, and ns for not significant).

**Figure 3 genes-12-00724-f003:**
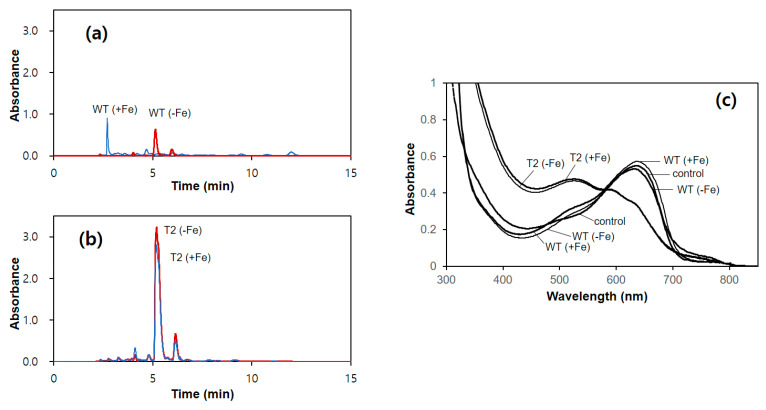
Production of siderophore by the wild-type and T2 strain. (**a**) HPLC analysis of the culture supernatant of the wild-type (WT) or the transformant (T2) grown in a minimal medium containing 10 μM Fe^3+^. (**b**) HPLC analysis of the culture supernatant of the wild type (WT) or the transformant (T2) grown in minimal medium without iron. (**c**) Modified chrome azurol-S (CAS) assay for the detection of siderophore. The peak fraction (5 min) from the HPLC analysis was subjected to a CAS assay.

**Figure 4 genes-12-00724-f004:**
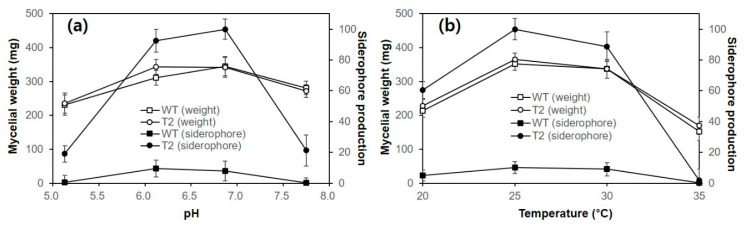
Effect of culture pH and temperature on the siderophore production and the mycelial growth of the wild-type (WT) and the transformant (T2). (**a**) Effect of pH. The strains were grown in C-PDB (50 mL) with different pH values for 2 weeks at 25 °C. (**b**) Effect of temperature. The strains were grown in C-PDB (50 mL) for 2 weeks at different temperatures. Siderophore in the culture supernatant was analyzed by a CAS assay, and the mycelial mass was measured after drying at 60 °C for 24 h. The siderophore productions shown here are the relative productions compared to the maximum values under the experimental conditions. Both experiments were carried out in triplicate, and the mean values are plotted with standard errors as error bars.

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
