# Peer review of "Generation of Iron-Independent Siderophore-Producing Agaricus bisporus through the Constitutive Expression of hapX"

_genes, 2021, doi:10.3390/genes12050724_

Round 1
Reviewer 1 Report
This manuscript contains a successful overproduction of siderophores in genetically modified commercial mushroom, A. bisporus. The experiments are well designed and methods are described in detail. The results were clearly stated and discussed concisely. I am sure that this work is of interest in many readers especially who are spending much efforts to establish molecular breeding in mushrooms, cause basidiomycetes are usually using different gene expression systems compared to well-studied ascomycete fungi.
I have one minor question about karyotype of the transformants. The authors stated that some transformants contained more than one ectopic hapx genes, as well as the endogenous hapx. But do you know if there is only one nuclei or several nucleus with different number of hapx genes exist in a single mycelial cell? I think that A. agaricus is basically homokaryon. But is it as well in the transformants, in regard with ectopically integrated genes? If you have some data, please describe about them in the results. If not, it could be at least mentioned in the discussion. I feel that this might be related to the fact that more than two ectopic hapx were observed in this work, besides single copy integration is common in ATMT of other organisms.
Some other points to be considered are as follows:
L.215: 'more than one' is suitable here? Maybe 'not less than one' or 'at least' seems better to me.
L.314: delete one of the 'well'.
That's about it from me.
Author Response
Dear Reviewer,
I'd like to express my sincere thanks to your helpful comments and suggestions. I hope the revision reflects and answers all of your concerns. Please go over following responses with pdf file of revised manuscript.
Sincerely,
HS Ro
Reviewer 1: Comments and Suggestions for Authors
1) I have one minor question about karyotype of the transformants. The authors stated that some transformants contained more than one ectopic hapx genes, as well as the endogenous hapx. But do you know if there is only one nuclei or several nucleus with different number of hapx genes exist in a single mycelial cell? I think that A. agaricus is basically homokaryon. But is it as well in the transformants, in regard with ectopically integrated genes? If you have some data, please describe about them in the results. If not, it could be at least mentioned in the discussion. I feel that this might be related to the fact that more than two ectopic hapx were observed in this work, besides single copy integration is common in ATMT of other organisms.
Response: We totally agree on your view on the multicopy integration observed in certain transformants. We currently don’t have any experimental data on this issue that requires elaborate analysis on basidiospores. However, we put following new lines in the discussion in Lines 344-347, as your suggestion.
“The multiple copies observed in certain transformants in this study do not imply multiple integration to the chromosomes of a certain nucleus because A. bisporus can have multiple nuclei in the cytoplasm.”
2) L.215: 'more than one' is suitable here? Maybe 'not less than one' or 'at least' seems better to me.
Response: Corrected as “The transformants carried at least one additional copy of hapX gene.” (L228)
3) L.314: delete one of the 'well'.
Response: removed. Thanks. (L329)

Reviewer 2 Report
This manuscript reports the over-expression of hapX in Agaricus bisporus and shows an increase in an unspecified siderophore in the transformants, with some link between gene dosage and siderophore expression level. There are few report of genetically manipulating Agaricus and none that I know of for altering its siderophores, so this is novel research. The manuscript is largely well written.
Specific comments foe consideration/alteration:
In the abstract and intro it is unclear which statements are derived from model species and which are specific to Agaricus. For example line 8-9 states siderophore production is only possible in an iron-free minimal medium since iron represses hapX... To my knowledge this has never been investigated in Agaricus, and is inferred from observations in the better studied ascomycetes, and indeed is only true the secreted siderophores, not those used intracellularly for iron storage (which are sometimes still expressed under iron replete conditions). These sorts of statements need to be clarified as they are misleading.
line 14 spelling mistake in A. bisporus
lines 38-59 these are quite general comments, and don't mention the known siderophores from Agaricus, or indeed from other model basidiomycetes such as Coprinopsis cinerea. I'd have expected the introduction to be far more specific to basidiomycete content.
Line 60-65, the knowledge about hapX is from which species? To my knowledge this hasn't been tested in any basidiomycete, and often transcription factors differ between asco- and basidiomycetes.
methods: line 96, how was the hapX gene identified? The stated accession number is an unannotated hypothetical gene, with no explanation as to why this was selected as being HapX. There are a number of bZip transcription factors in the genome, its not clear why this was the one selected.
line 179-187, how was the Sep-Pak purification performed - volume of media processed, washed with what, eluted with what? HPLC conditions are not described - what solvent system, was it a gradient? Did the authors also look for mycelial siderophores?
Were the samples flooded with iron to ensure all siderophores are in the iron-bound state, if not, there would typically be an additional peak for each desferri-siderophore on HPLC.
I found the details of the CAS assay unclear, there are are at least three known siderophores from Agaricus, it is unclear to me which is being seen in the HPLC and whether the CAS assay is measuring just this one, or a combination of all siderophores present.
I am happy with all the data on obtaining and initial molecular analysis of the transformants - this was nicely presented.
lines 250-257, please chack what you mean by copy number as this is unclear in places. I assume you have dikaryon material (so actually two copies rather than one in the wild type). You then talk about toaltl copies or sometimes additional copies and this should be more consistent. Especially in line 256 where you say T1 and T4 both have three copies, this is not correct. T1 has two additional transgenic copies by T4 only has one additional transgenic copy.
Where is the data for sidC - you ccomment on this in the discussion but results were not presented?
line 269- see above about whether extracts were soaked with Fe3+ before HPLC, the exracts from iron-free media should have a different retention time from those in iron-containing media if not pre-soaked with iron prior to HPLC. There are three secreted siderophores known for Agaricus under iron starvations, which is being seen here and where are the others?
line 289 - the fungi were grown on C-PDB which will contain iron, so the wild-type would be expected to have little or no siderophore production, I'm unclear as to the relevance of this and of the data shown in fig4.
line 307surely all mushrooms belong to the basidiomycota, not "most".
line 312, the statement that there have been few studies on genetic engineering of basidiomycetes seems to ignore the enormous amount of information from Coprinopsis cinerea, or indeed Phanerochaete chrysosporium or Schizophylum commune.
line 338, the detail about known Agaricus siderophores ought to have been in the introduction, and ideally this study should have confirmed which siderophores are being detected (MS/nmr).
Author Response
Dear reviewer,
I deeply appreciate your comprehensive and helpful comments and suggestion to improve our manuscript. I hope the revised manuscript answers all of your concerns.
Sincerley,
HS Ro
Reviewer 2: Comments and Suggestions for Authors
1) In the abstract and intro it is unclear which statements are derived from model species and which are specific to Agaricus. For example line 8-9 states siderophore production is only possible in an iron-free minimal medium since iron represses hapX... To my knowledge this has never been investigated in Agaricus, and is inferred from observations in the better studied ascomycetes, and indeed is only true the secreted siderophores, not those used intracellularly for iron storage (which are sometimes still expressed under iron replete conditions). These sorts of statements need to be clarified as they are misleading.
Response: We observed repression of hapX and inhibition of siderophore production both in the complex medium and iron supplemented-minimal medium in Agaricus bisporus as shown in Figs. 2 and 3. For the clarity we changed the line 8-9, and line 73 as follows:
Line 8-10: “Siderophore secretion in A. bisporus was enabled only in iron-free minimal medium due to iron repression of hapX, a transcriptional activator of siderophore biosynthetic genes.”
Line 78:” However, A. bisporus, like other fungi, secreted siderophores only in the absence of iron, meaning that the production medium should be an iron-free minimal medium.”
2) line 14 spelling mistake in A. bisporus
Response: Corrected. Thanks.
3) lines 38-59 these are quite general comments, and don't mention the known siderophores from Agaricus, or indeed from other model basidiomycetes such as Coprinopsis cinerea. I'd have expected the introduction to be far more specific to basidiomycete content.
Response: We agree on your comments. The detailed descriptions on the biosynthetic process of siderophores are replaced with new paragraphs focusing on NRPS in basidiomycetes(line 43-55 in the revised manuscript).
4) Line 60-65, the knowledge about hapX is from which species? To my knowledge this hasn't been tested in any basidiomycete, and often transcription factors differ between asco- and basidiomycetes.
Response: The knowledge about hapX mostly comes from ascomycetes indeed. However, hapX in Cryptococcus neoformans, a basidiomycete yeast, has been extensively studied that demonstrates the hapX functions in both the ascomycetes and basidiomycetes are homologous. We added new lines on this in the revised manuscript line 61-64 with two new citations.
Jung etal. HapX positively and negatively regulates the transcriptional response to iron deprivation in Cryptococcus neoformans. PLoS Pathog. 2010, 6, e1001209.
Do et al. A transcriptional regulatory map of iron homeostasis reveals a new control circuit for capsule formation in Cryptococcus neoformans. Genetics 2020, 215, 1171-1189..
5) methods: line 96, how was the hapX gene identified? The stated accession number is an unannotated hypothetical gene, with no explanation as to why this was selected as being HapX. There are a number of bZip transcription factors in the genome, its not clear why this was the one selected.
Response: The hapX gene in A. bisporus was retrieved from the A. bisporus genome information in MycoCosm DB using Aspergillus fumigatus HapX protein as query sequence. Domain analysis of the retrieved sequence showed the presence of Hap2/3/5 binding domain and bZIP domain similar to A. fumigatus HapX protein. The HapX was also present in other basidiomycetes such as Pleurotus eryngii and P. ostreatus. To explain this we added new description in the method section, line 95-102, with supplementary Figure S1. We also added information on A. bisporus sidC and sidD.
6) line 179-187, how was the Sep-Pak purification performed - volume of media processed, washed with what, eluted with what? HPLC conditions are not described - what solvent system, was it a gradient? Did the authors also look for mycelial siderophores?
Response: We added detailed experimental procedure in line 191-200 in the revised manuscript. On the mycelial siderophore, we did not check. We only focused on the extracellular siderophores.
7) Were the samples flooded with iron to ensure all siderophores are in the iron-bound state, if not, there would typically be an additional peak for each desferri-siderophore on HPLC.
Response: Yes. We added 1 mM FeCl3 before the Sep-Pak extraction. (Line 191-192).
8) I found the details of the CAS assay unclear, there are at least three known siderophores from Agaricus, it is unclear to me which is being seen in the HPLC and whether the CAS assay is measuring just this one, or a combination of all siderophores present.
Response: As your comments, A. bisporus has been known to produce at least three siderophores. However, we currently have no information on the nature of the siderophore that we obtained from the transformant. We just have a guess that it is fusarinine C or TAFC because it was produced accompanied by SidD expression not SidC.
I am happy with all the data on obtaining and initial molecular analysis of the transformants - this was nicely presented.
9) lines 250-257, please chack what you mean by copy number as this is unclear in places. I assume you have dikaryon material (so actually two copies rather than one in the wild type). You then talk about toaltl copies or sometimes additional copies and this should be more consistent. Especially in line 256 where you say T1 and T4 both have three copies, this is not correct. T1 has two additional transgenic copies by T4 only has one additional transgenic copy.
Response: You are right. We made mistake. We corrected as follows (L266-270):
“ The transformant T2, which contained three additional transgenic copies of the hapX gene, showed only 1.3-fold more hapX gene expression than T15, which had two additional copies of the hapX gene. Moreover, the expression level of sidD in T2 was almost the same as that in T15. Notably, the sidD expression in T1, which have two additional hapX copies, was lower than that in T15.”
10) Where is the data for sidC - you ccomment on this in the discussion but results were not presented?
Response: We indeed check the expression of sidC and sidD through RT-PCR. However, we were only able to detect the expression of sidD. We marked the comments on sidC in line 363 “data not shown”.
11) line 269- see above about whether extracts were soaked with Fe3+ before HPLC, the exracts from iron-free media should have a different retention time from those in iron-containing media if not pre-soaked with iron prior to HPLC. There are three secreted siderophores known for Agaricus under iron starvations, which is being seen here and where are the others?
Response: Samples were pretreated with FeCl3 as explained in 7). The siderophore produced in iron-starvation in WT appears to be the same with that produced by the transformant. As explained in 8), we just guess that it is fusarinine C or TAFC because it was produced accompanied by SidD expression not SidC.
12) line 289 - the fungi were grown on C-PDB which will contain iron, so the wild-type would be expected to have little or no siderophore production, I'm unclear as to the relevance of this and of the data shown in fig4.
Response: We think iron-chelating metabolites, such as oxalate and citrate, can make such noise in CAS assay. We added line 310-312 to explain this.
“It is notable that the iron-chelating activity observed in WT may come from iron-chelating metabolites, such as oxalate and citrate.”
13) line 307surely all mushrooms belong to the basidiomycota, not "most".
Response: Fruiting bodies (ascocarps) of some ascomycetes, such as morel, are sometimes called as mushrooms and that is why we reluctantly inserted “most”.
14) line 312, the statement that there have been few studies on genetic engineering of basidiomycetes seems to ignore the enormous amount of information from Coprinopsis cinerea, or indeed Phanerochaete chrysosporium or Schizophylum commune.
Response: Please understand that the description here is to emphasize scarcity of genetic engineering in the edible and medicinal mushroom side.
15) line 338, the detail about known Agaricus siderophores ought to have been in the introduction, and ideally this study should have confirmed which siderophores are being detected (MS/nmr).
Response: Yes, we added this in the introduction in the revised version. Please understand that the identification of siderophore is out of our capacity for now.

Round 2
Reviewer 2 Report
All aspects have been addressed appropriately